# Informing children of their parent's illness: A systematic review of intervention programs with child outcomes in all health care settings globally from inception to 2019

Charlotte Oja[1], Tobias Edbom[2], Anna Nager[1], Jörgen Månsson[3], Solvig Ekblad [4,5]*

1 Department of Neurobiology, Care Sciences and Society (NVS), Division of Family Medicine and Primary Care, Karolinska Institutet, Stockholm, Sweden, 2 Department of Clinical Neuroscience (CNS), Center for Psychiatric Research, Karolinska University Hospital, Stockholm, Sweden, 3 Department of Public Health and Community Medicine/Primary Health Care, Institute of Medicine, Sahlgrenska Academy, University of Gothenburg, Gothenburg, Sweden, 4 Academic Primary Health Care Center, Region Stockholm, Stockholm, Sweden, 5 Department of Learning, Informatics, Management and Ethics (LIME), Karolinska Institutet, Stockholm, Sweden

* Solvig.Ekblad@ki.se

**Data Availability Statement:** All relevant data are within the manuscript and its Supporting Information files.

## Abstract

### Introduction

Children are impacted when parents are ill. This systematic review gives an overview of the current state of research and extracts what children and parents found helpful in the interventions aimed at informing children of their parent's illness.

### Methods

This review was registered with PROSPERO and conducted in accordance with PRISMA guidelines. Five health and social science databases were searched from inception to November 2019 to identify original, peer-reviewed articles in English describing effective interventions. The authors selected and reviewed the studies independently, and any inconsistencies were resolved by discussion in face-to-face meetings and emails. A descriptive synthesis of evidence-based concepts from quantitative and qualitative studies was conducted.

### Results

A total of 13 892 titles and 144 full-text articles were reviewed with 32 selected for final inclusion, 21 quantitative, 11 qualitative and no mixed-method studies published from 1993 to November 2019. Most of the research was conducted in mental health, including substance abuse (n = 22), but also in cancer care (n = 6) and HIV care (n = 4). Most studies using quantitative method showed a small to moderately positive statistically significant intervention effect on the child's level of internalized symptoms. Content analysis of the results of studies employing qualitative methodology resulted in four concepts important to both children and parents in interventions (increased knowledge, more open communication, new coping

**Funding:** The funder Capio (https://capio.com/en/) provided support in the form of salary for only the author Charlotte Oja but did not have any additional role in the study design, data collection and analysis, decision to publish, or preparation of the manuscript. The specific roles of the authors are articulated in the 'author contributions' section.

**Competing interests:** Charlotte Oja is employed by Capio as a clinical Medical Doctor, Specialist of General Practice. This does not alter our adherence to PLOS ONE policies on sharing data and materials.

strategies and changed feelings) and three additional concepts important to parents (observed changes in their children's behavior, the parent's increased understanding of their own child and the relief of respite).

## Conclusions

In the literature there is evidence of mild to moderate positive effects on the child's level of internalized symptoms as well as concepts important to children and parent's worth noting when trying to bridge the still existing knowledge gaps. In further efforts the challenges of implementation as well as adaptation to differing clinical and personal situations appear key to address.

## Introduction

### Transgenerational transmission of illness

Research in human development is, according to The Ecological Systems Theory, recommended to focus on the progressive accommodation, throughout the life span, between the growing human organism and the changing environments in which it lives and grows [1]. In this broad sense, parental illness is seen as a risk factor for current and future ill health in the children of the family. For instance, a systematic review [2] found an association between depression in mothers and depression in their school-aged children. Also, the presence of governmental economic support to sick parents is associated with an elevated risk of health-related inability to work for their children when the children become young adults [3]. Paths of transmission have been shown to include genetic factors, shared social and economic factors and the illness' effect on the parenting function [4]. In the child psychology literature, the parenting function includes: emotional connectedness, attachment and care ability [5], routines, family values and climate, social and economic stability, safety, and support for the child to develop and gradually become appropriately independent. Based on a large number of relevant theoretical reviews of child psychological development, Diareme [6] presents children's developmental issues related to somatic parental illness as follows:

- *Infancy*: Separation from parent(s) and inconsistent physical and emotional care by parent(s);

- *Toddlerhood*: Separation from parent(s) (experienced as abandonment or punishment) and inconsistent provision of attention and limit setting;

- *Preschool*: Magical thinking of having caused the parent's illness, illness experienced as punishment, and fun and play perceived as inappropriate;

- *Latency*: Irrational fear of causing or exacerbating parent's illness and associated guilt, somatic complaints (due to age-expected identification with ill parent), fear of losing the healthy parent (due to age-expected dependency on parents), guilt for having fun, and feeling unimportant;

- *Adolescence*: Guilt or ambivalence about desire for independence (due to age-expected need for autonomy), somatic complaints (due to age-expected concerns with body image and health identity formation issues), shame about ill parent (due to age-expected need for peer acceptance), resentment of increased responsibilities at home (due to age-expected need for

independent activities), negligence or compromise of own growth and autonomy (due to age-expected guilt for wanting to separate from ill parent) [6].

At any age the child can respond to parental low mood in different ways, functional or dysfunctional. Four patterns of child response have been discovered: Active Empathy, Emotional Over-involvement, Indifference and Avoidance. The Emotional Over-involvement and Avoidance groups reported more depressive and externalized symptoms than the other two groups [7].

## Law and policy protecting and supporting child development

Also, from a perspective of law and policy there are strong reasons to focus on protecting the health of the growing child. The Global Goals of the UN's 2030 Agenda for Sustainable Development challenge us all in Goal Number Three to "Ensure healthy lives and promote well-being for all at all ages" [8]. The United Nations Convention on the Rights of the Child [9], which was a Swedish law from January 1, 2020, states that the child should enjoy rights including the right to be listened to and at the same time be given special protection, opportunities and facilities. In addition, some nations have national regulations guiding health care staff of adult patients to attend to needs of these patients' children. For example, in Sweden health care personnel are required by law [10] to give information, advice and support to under aged children whose parents are seriously psychiatrically or physically ill, have an addictive disorder (including gambling), or have died.

## Interventions aimed at reducing the impact of the parental illness on the child

The widespread acknowledgement that children in families affected by parental illness are at risk for a range of poor life outcomes has resulted in the creation of interventions aimed at reducing the negative impact of the parental illness on the child and supporting the healthy development of the child.

**Somatic parental illness.** Interventions aimed at somatic parental illness were reviewed in 2007 [6]; the review presented an overview of published studies, organized according to type of intervention, children's age, stage of parental illness, and the goals, techniques, theoretical basis, duration, evaluation, and outcome of the intervention. It concluded that most of the published studies concerned school-age children and that the interventions provided support and illness-related information primarily in the context of short-term group interventions. A systematic review in 2017 of children's psychosocial needs and existing interventions for children facing a parent's cancer diagnosis [11] concluded that: (1) Children need age-appropriate information about their parent's cancer; (2) Children require support communicating with parents, family members and health professionals; and (3) Children need an environment where they feel comfortable sharing positive/negative emotions and can have their experiences normalized among peers. Another systematic review [12] analyzed barriers and facilitating factors to the implementation and use of interventions. Barriers to the use of support services were practical difficulties, perceived need for support, fear of stigma, disease characteristics and complications in collaborations. Additionally, intervention characteristics such as being perceived as routinely offered, with a flexible structure and accessibility, were important. In the specific case of parental incurable end-stage cancer, a rapid evidence assessment revealed four studies [13] and concluded that targeted, child-centered, family-focused psychosocial interventions were sometimes used but that there was a need for comparative effectiveness studies that test the timing, delivery, and content of these interventions. A review of

interventions in the case of physical illness other than HIV and cancer [14] identified nine support interventions, and most were pilot studies. The review showed that the main aim of most interventions is to enhance family functioning for all afflicted families by helping parents to communicate with their children. A review of intervention programs for children of parents with Multiple Sclerosis showed that those children who increase their coping skills, social support, and knowledge of the disease process exhibit decreased emotional distress and increased overall life satisfaction [15]. A review of interventions for family members of Intensive Care Unit patients identified the use of support groups for family members of patients, structured communication and/or education programs for family members, the use of leaflets or brochures to meet family information needs, the use of a diary, changes in the physical environment, and open or more flexible visiting hours as effective [16].

**Mental parental illness.**   In 2006 a critical review of intervention programs for children with parents with mental illness [17] concluded that there was very limited evidence of program effectiveness in terms of improved well-being of the child. Encouragingly, in 2017 a systematic review and meta-analysis of randomized controlled trials [18] of pedagogical intervention programs for children of depressed parents showed that the increased risk of childhood depression could be partly mitigated. A meta-analysis in 2017 [19] reported on 96 articles, including 50 independent samples from randomized controlled trials quantifying effects of preventive interventions. Random effect models resulted in small though significant Effect Sizes for programs enhancing mother-infant interaction as well as mothers' and children's behavior. Intervention programs for children/adolescents resulted in statistically significant small positive effects for global psychopathology as well as internalizing symptoms and this positive effect increased over time. Externalized symptoms also reached significance in the follow-up assessments. Interventions addressing parents and children jointly produced overall larger effects. Another review in 2017 [20] concluded that there was an emerging evidence base demonstrating that including parenting as a focus of recovery practice is effective in improving parental, child and family well-being.

**Substance abuse.**   A comprehensive systematic review [21] of selective prevention programs for children from substance-affected families showed preliminary evidence for the effectiveness of the programs, especially when their duration was longer than ten weeks and when they involved children's, parenting, and family skills training components. Outcomes proximal to the intervention, such as program-related knowledge, coping-skills, and family relations, showed better results than more distal outcomes such as self-worth and substance use initiation.

## The influence of different settings

Health care is provided in many different settings. The characteristics of each setting strongly influence the possibility and appropriateness of delivering interventions. Every specific clinical encounter is set in a specific situation defined by multiple factors. For instance, is it a local clinic or a hospital setting? Is the parental illness psychiatric, somatic, addictive or multi-morbid? Is the parental illness non-life threatening or life threatening? Is the parental illness in an acute, chronic or terminal stage?

**Primary health care.**   Primary Health Care tends to the needs of most patients both in early and late stages of disease investigation and management. All types of diseases are diagnosed and treated and multimorbidity is common. Opportunities for family practitioners to assist families with serious parental mental illness were described nearly 30 years ago [22]. The potential described seems from the literature still largely unfulfilled. In one study Norwegian adolescents, each with an ill or substance-abusing parent, consulted GPs mostly for somatic

complaints. They wanted to be met both as a unique person and as a member of a family with burdens. Their expectations from the GP were partly negatively formed by their experiences. Some had experienced that both their own and their parent's health problems were not addressed properly. Others reported that the GP did not act when he or she should have been concerned about their adverse life situation [23]. Generally, there was a knowledge gap in research and development in this most common health care setting. We were therefore specifically interested in finding interventions tested in Primary Health Care. If there were few studies tested in a primary health care setting, the need to learn about effective components of interventions in other health care settings was apparent and urgent. Interventions based on these learnings can then be adapted, and their effectiveness tested in a variety of primary health care settings.

## Aims and significance of this review

This systematic review aims to identify evidence-based/ evidence-informed concepts found in effective interventions aimed at informing children of their parent's illness in all health care settings globally. The significance was to gather and extract universally applicable "good practice" experience which are helpful when attempting to choose, tailor and adapt interventions for different health care settings, including primary health care. The following research questions were investigated.

## Research questions

1. What is known about contextual factors, such as type of health care setting, type of parental illness (somatic, mental, substance abuse), stage of illness (acute, chronic, terminal), type of intervention (individual, family, peer group), and cultural context?

2. What evidence is available regarding effective intervention programmes on the outcomes of internalized and externalized symptoms and psychosocial function of the child?

3. What are some evidence-based effective components in the intervention?

## Methods

This review involved structured searches of peer-reviewed literature in five medical and social databases (Medline/PubMed (Ovid), Web of Science Core Collection, PsycInfo (Ovid), Cinahl and SveMed+) and was conducted in accordance with the Preferred Reporting Items for Systematic Reviews and Meta-Analyses (PRISMA) guidelines [24]. Details of the protocol for this systematic review were registered on PROSPERO (International Prospective Register of Systematic Reviews) and can be accessed at https://www.crd.york.ac.uk/PROSPERO/#recordDetails.

## Search strategy and eligibility criteria

As a first step, we identified a few key articles known to us [25–27] and through them and databases sought out relevant search terms.

In the second step, we searched without restrictions on language, year, or publication type in the following databases: Medline/PubMed (Ovid), Web of Science Core Collection, PsycInfo (Ovid), Cinahl, and SveMed+ to identify relevant articles and references. The searches were conducted by two librarians at the Karolinska Institutet University Library in January

2018 and November 2019, to update publications. The complete search strategies are available as a supplementary file in Appendix 1. The extensive search strategy included both free-text and MeSH terms and was initially created in Medline/PubMed and later adapted to the other databases with corresponding vocabularies.

All titles were screened by CO until 2018, and SE and AN each read half of the number of titles between 2018—November 2019, and all obviously irrelevant articles, such as those relating to giving parents information about the illness of their sick child, were excluded. All relevant abstracts were screened according to the inclusion criteria by CO until 2018 and SE and AN each read half of the number of the abstracts between 2018 and November 2019. Meanwhile until 2018, SE, JM, AN, and TE screened a fourth of the abstracts each which continued between 2018 and November 2019. The suggested inclusions were then compared, and if CO and the others agreed of the articles up to 2018, the article was included. Between 2018 – November 2019 AN and TE and SE and JM, respectively read the same articles, compared and agreed for the inclusion. If there was any uncertainty as to whether articles should be included or not, the articles in question were discussed in the full group (up to 2018, and AN, SE, TE and JM for articles between 2018 –November 2019) before a final decision was reached.

**Inclusion criteria.** Peer-reviewed articles, (excluding PhD reports, but published articles in such reports were included) were retained for the review if the study focused on:

- Intervention programs aimed at benefitting underaged children of sick parents

- A study population defined as patients, (possibly partners) and their children

- Intervention programs that could be carried out on individual or family or group levels

- Any parental medical condition (All parental medical conditions were included)

- Outcomes that were defined as qualitatively or quantitatively measured symptoms or effects on the child of the ill parent

**Exclusion criteria.** Articles were excluded if they:

- Presented a relevant field of research but no intervention program was reported

- Provided no qualitative or quantitative data on the effect of an intervention program on the child

- Reported intervention studies but without randomization

- Had no abstract in English, which later was changed to no main text in English, as the research group found itself unable to handle articles in German, French and Norwegian. We did not contact authors of Non-English articles due to time and monetary constraints.

- Focused on intervention programs based on specific social factors such as parental military service or imprisonment

- Focused on educational interventions aimed primarily at increasing professional skills

## Screening, selection and data extraction

Following an initial screening of titles and abstracts, 144 potentially relevant articles were selected for review. All 144 full text articles were independently screened by at least two authors, CO and either SE, TE, JM, or AN in 2018. Any uncertainty concerning inclusion or exclusion was resolved by discussion among all five authors in 2018. A total of 57 relevant

articles remained. Of these, 28 had a descriptive design with pre- and post-measurements but without randomization. These articles described valuable learning in the field of mental health (including substance abuse) n = 17 [28–44], cancer n = 6 [45–50], HIV n = 1 [51], Multiple Sclerosis n = 1 [52], acquired brain injury n = 1 [53], traumatic grief n = 1 [54], and critical care n = 1 [55] but were excluded because of less rigorous methodological design. The same process was done for publications between 2018 and November 2019 in the group of SE, AN, TE and JM. The reasons for exclusion were unrelated topic, no full text in English, not original peer-reviewed research, no clear outcome in the health of the child, and multiple reports from the same study (Fig 1).

Data were extracted on article publication characteristics (e.g. year of publication, journal), study procedures (e.g. study design, participant recruitment methods, data collection methods), participant characteristics (health care setting, parental illness, stage of illness, types of intervention, and country) and effects of intervention and documented in a purpose-designed Microsoft Excel sheet and independently reviewed by the four coauthors.

## Quality assessment of included articles

A quality assessment was conducted using Critical Appraisal Skills Programme (CASP) checklists to assess methodological quality of included studies [56]. The quality assessment focused on assessing the strengths and weaknesses of each study. A total score was calculated for each study based on relevant checklist items and each study was graded as being of low, moderate or high quality by CO and either SE, TE, JM, or AN. Any disagreements were resolved by discussion.

## Data synthesis

Due to the heterogeneity of study designs, contexts, populations and out-come measures, a meta-analytical approach was not considered appropriate. A narrative synthesis in four steps was conducted, drawing on the framework and techniques described in 'ERSC Guidance on Conducting Narrative Synthesis' [57]. In accordance with the guidance, a model of change [58] was created as a first step at the outset of data synthesis. This model presents how the interventions work in a general and broad sense, why they work, and for whom (Fig 2).

In a second step a preliminary synthesis was developed to organize the findings from included studies to describe patterns across the studies in terms of direction and size of effects. Initially, the quantitative and qualitative studies were synthesized separately.

Tabulated quantitative data were reorganized in groups according to type of parental illness. The intervention outcome on included children internalized and externalized symptoms and pro-social function was tabulated as presented in the original articles. Articles reporting statistically significant improvement of the child's symptoms were categorized as "improved".

Tabulated qualitative data were reorganized in groups according to type of parental illness. Intervention outcomes for the children and parents, respectively, were tabulated in the wording of the original papers. These intervention outcomes for children and parents were extracted and imported into NVivo, where a content analysis [59] was conducted.

In a third step relationships in and between studies were explored. The results from the qualitative studies in cancer settings were analyzed separately and then compared to an analysis of the results from studies in mental health settings.

In a fourth step the robustness [57] of the analysis was assessed. The quality of the primary included studies is high, as all others were excluded, leaving the analysis with only 22% of the initially relevant articles. The analysis is trustworthy, as all authors have checked that the transferal of information from the original article texts is correct, and all have confirmed and after

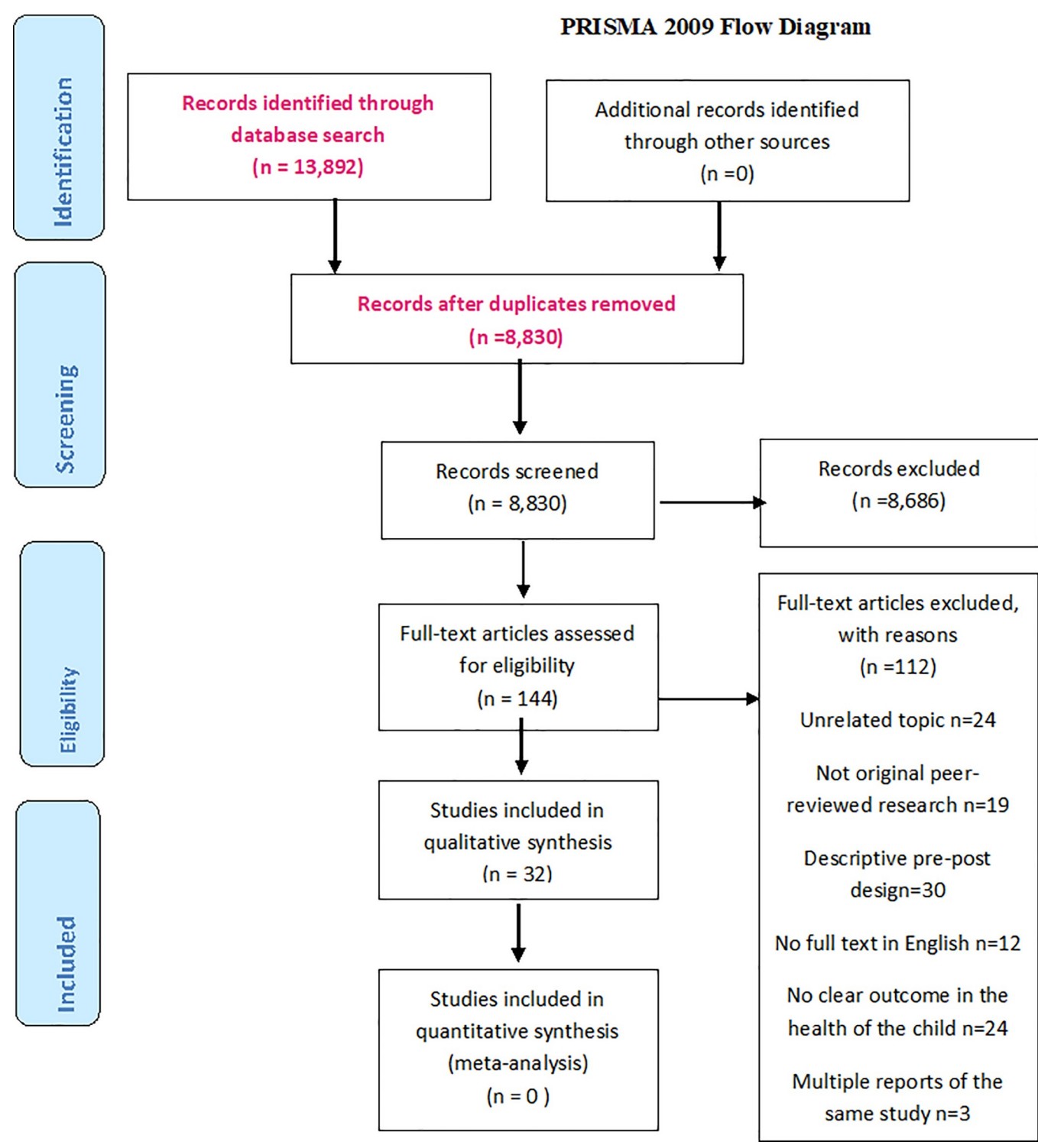

**Fig 1. Flow chart of articles screened and selected for review published until November 2019.**

discussion agreed on the result of the analysis. In May 2019 the results of the qualitative articles were analyzed according to an established method, i.e. content analysis.

Intervention where children are informed about parental illness in an appropriate way

Child's needs addressed

Parental needs addressed

Safety

Growth

Intact parental role

Increased SOC

Low stress

Growth from life challenge

Retained self-esteem

Lower stress

Less symptoms

Increased capacity

Parents have good relations with child

Child develops self-worth and capacity

Child is healthier

**Fig 2. Programme theory model: Mechanism by which interventions facilitate child's health.**

## Results

### Study characteristics

A total of 32 articles were included in the review. The methods used were either quantitative or qualitative, and the resulting articles are presented by the methods used below in Tables 1 and 2. The wording in the tables follows the original authors' terminology when possible.

### Study types and quality assessment

Included were 21 quantitative, 11 qualitative and no mixed-methods studies published from 1994 to November 2019. Of them, 18 studies were rated as having high methodological quality (56%). Most of the research was conducted in mental health, including substance abuse (n = 22), but there were also studies done in cancer care (n = 6) and HIV care (n = 4).

### Synthesis of findings

### Quantitative methodology studies

**Contextual factors in studies using quantitative methodology.** Fifteen of the 21 studies with quantitative design were conducted with families with parental mental health problems, 4 with families with parental HIV and 2 with parental cancer. Mental health studies mostly concerned parental depression, but there were also studies in the context of parental anxiety, bi-polar disorder, ADHD and drug abuse. The children were in the age span of 4 to 17 years, except for one study focusing on infants of drug-dependent mothers. The follow-up times range from one to four years. All studies concern chronic parental disease except two, where parents were terminal with either cancer or HIV. The studies were conducted predominantly in the USA and Europe, but there was also one study each from China, Myanmar and South Africa.

**Intervention programme effects on the outcomes internalizing and externalizing symptoms and psychosocial function of the child.** *Child outcome, internalized symptoms.* Eighteen of 20 studies showed a small to moderately positive statistically significant intervention effect on the child's level of internalized symptoms; two showed no effect. The study on infants showed marginally higher cognitive scores.

**Table 1. Included studies using quantitative design.**

| Reference | Year conducted | Title | Journal | Population | Intervention | Child out-come | Parental illness | Illness stage | Age of children (yrs) | Follow up (mths) | Setting | Location | Child out-come internalizing symptoms | Child out-come externalizing symptoms | Child outcome Psycho-social function | Quality assesment |
|---|---|---|---|---|---|---|---|---|---|---|---|---|---|---|---|---|
| van Santvoort, F. H., C. M.; van Doesum, K. T.; Janssens, J.M. | 2014 | Effectiveness of preventive support groups for children of mentally ill or addicted parents: a randomized controlled trial | European Child & Adolescent Psychiatry | Children with info to parents | Peer support group, parental talk, concluding meeting with family | SDQ | Mental health | Chronic | 8 to 12 | 12 months | Psychiatry | The Netherlands | Improved. | No effect. | Improved. | High |
| Solantaus, T. P., E. J.; Toikka, S.; Punamaki, R. L. | 2010 | Preventive interventions in families with parental depression: children's psychosocial symptoms and prosocial behaviour | European Child & Adolescent Psychiatry | Family | Beardslees family intervention vs Let's talk about children | SDQ | Mental health | Chronic | 8 to16 | 4, 10 and 18 months | Psychiatry | Finland | Improved. | Improved. | Improved. | High |
| Punamaki, R. L. P., J.; Toikka, S.; Solantaus, T. | 2013 | Effectiveness of preventive family intervention in improving cognitive attributions among children of depressed parents: a randomized study | Journal of Family Psychology | Family | Beardleeds family intervention vs Let's talk about children | CDI/BDI, SDQ, CASQ-R | Mental health, affective disorder | Chronic | 8 to 16 | 10 and 18 months | Psychiatry | Finland | Improved. | | | High |
| Ginsburg, G. S. D., K. L.; Tein, J Y.; Teetsel, R.; Riddle, M. A. | 2015 | Preventing Onset of Anxiety Disorders in Offspring of Anxious Parents: A Randomized Controlled Trial of a Family-Based Intervention | American Journal of Psychiatry | Family | Family intervention, The Coping and Promoting Strength intervention vs information-monitoring control condition. | ADISC | Mental health, anxiety | Chronic | 6 to 13 | 12 months | Psychiatry | USA | Improved. | | | High |
| Garber, J. C., G. N.; Weersing, V. R.; Beardslee, W. R.; et al | 2009 | Prevention of depression in at-risk adolescents: a randomized controlled trial | JAMA | Peer group with info to parents | Group Cognitive Behvioral Intervention vs usual care | CES-D | Mental health, depression | Chronic | 13 to 17 | 6 months | Psychiatry | USA | Improved. | | | High |
| Compas, B. E. F., R.; Thigpen, J.; Hardcastle, E.; et al | 2015 | Efficacy and moderators of a family group cognitive-behavioral intervention for children of parents with depression | Journal of Consulting & Clinical Psychology | Family group of four families vs written information mailed three times to families | Family Group Cognitive Behaviour intervention. | CES-D, CBCL, Child diagnostic interview | Mental health, depression | Chronic | 9 to 15 | 2, 6,12,18 and 24 months | Psychiatry | USA | Improved. | Improved. | | High |
| Beardslee, W. R. G., T. R.; Wright, E. J.; Cooper, A. B. | 2003 | A family-based approach to the prevention of depressive symptoms in children at risk: evidence of parental and child change | Pediatrics | Family | Beardsleeds family intervention (clinician facilitated vs lecture) | Interview of mother and child using Kiddie-SADS-E-R, YSR, Child interview | Mental health, depression | Chronic | 8 to 15 | 2.5 years | Psychiatry | USA | Improved. | | | High |
| Compas, B. E. F., R.; Thigpen, J. C.; Keller, G.; et al | 2011 | Family group cognitive-behavioral preventive intervention for families of depressed parents: 18- and 24-month outcomes | Journal of Consulting & Clinical Psychology | Family group | Family group cognitive behavioral (FGCB) preventive intervention | CES-D, CBCL, Child diagnostic interview | Mental health, depression | Chronic | 9 to 15 | 18 and 24 months | Psychiatry | USA | Improved. | Improved. | | High |
| Compas, B. E. F., R.; Keller, G.; Champion, et al | 2009 | Randomized controlled trial of a family cognitive-behavioral preventive intervention for children of depressed parents | Journal of Consulting & Clinical Psychology | Family group | Family group cognitive–behavioral (FGCB) preventive intervention | CES-D, CBCL, Child diagnostic interview | Mental health, depression | Chronic | 9 to 15 years | 2, 4, 6 and 12 months | Psychiatry | USA | Improved. | No effect. | | High |
| Jones, S. C., R.; Sanders, M.; Diggle, P.; et al | 2015 | A pilot web based positive parenting intervention to help bipolar parents to improve perceived parenting skills and child outcomes Addendum | Behavioural and Cognitive Psychotherapy | Parents | Web-intervention | SDQ, The Parenting Scale | Mental disorder, Bipolar disorder | Chronic | 4 to 10 | 10 weeks, post course | Internet | UK | Improved. | Improved. | | Moderate |

*(Continued)*

**Table 1.** (Continued)

| Reference | Year conducted | Title | Journal | Population | Intervention | Child out-come | Parental illness | Illness stage | Age of children (yrs) | Follow up (mths) | Setting | Location | Child out-come internalizing symptoms | Child out-come externalizing symptoms | Child out-come Psycho-social function | Quality assesment |
|---|---|---|---|---|---|---|---|---|---|---|---|---|---|---|---|---|
| Black, M. M. N., P., Kight, C.; Wachtel, R; et al | 1994 | Parenting and early development among children of drug-abusing women: effects of home intervention | Pediatrics | Mothers and infants | Home intervention vs care as usual | Bayley Scales of Infant Development, HOME, CAPI, Parenting stress index. | Mental health, drug-abuse | Chronic | 0 to 18 months | 18 months | Home-intervention | USA | At six months the infants obtained marginally higher cognitive scores | | Improved. | Moderate |
| Christ, G. H. R., V. H.; Seigel, K.; Karus, D.; et al | 2005 | Evaluation of a preventive intervention for bereaved children | Journal of Social Work In End-Of-Life & Palliative Care | Healty parent and children | Psycho-educational intervention targeting children through the healthy parent | CDI, SEI, STAIC, STAIY, POPM | Cancer,terminal, one parent dies | Terminal, incurable | 7 to 17 | Initial, 8 and 14 months after the parent's death | Oncology | USA | Improved. | No effect. | Improved. | Moderate |
| Mon, M. M. L., T.; Htut, K. M. | 2016 | Effectiveness of Mindfulness Intervention on Psychological Behaviors Among Adolescents With Parental HIV Infection: A Group-Randomized Controlled Trial | Asia-Pacific Journal of Public Health | Children with information to parents | Monthly mindfulness group sessions led by an experienced mindfulness trainer | SDQ | HIV | Mixed parental HIV stage | 10 to 16 | Six months after intervention | Township | Myanmar | Improved. | Improved. | | High |
| McKee, L. G. P., J.; Forehand, R.; Rakow, A.; Watson, K., et al | 2014 | Reducing youth internalizing symptoms: effects of a family-based preventive intervention on parental guilt induction and youth cognitive style | Development & Psychopathology | Parents and children | Family group cognitive–behavioral (FGCB) preventive intervention | YSR (11–18), CBCL, ACSQ, MGI | Mental health, depression | Chronic | 9 to 15 | 6, 12 and 18 months | Psychiatry | USA | Improved. | | | Moderate |
| Eloff, I. F., M.; Makin, J. D.; Boeving-Allen, A.; et al | 2014 | A randomized clinical trial of an intervention to promote resilience in young children of HIV-positive mothers in South Africa | AIDS | Mothers and children | Intervention groups with first mothers and children participating separately and thereafter together | CBCL, CDI | HIV | Chronic | 6 to 10 | 6, 12 and 18 months | Two separate communities within Tshwane (formerly Pretoria) | South Africa | No effect. | Improved. | Improved. | High |
| Rotheram-Borus, M. J. L., M.; Leonard, N.; et al | 2003 | Four-year behavioral outcomes of an intervention for parents living with HIV and their adolescent children | AIDS | Parent and adolescent children | Parental group 8 sessions, then adolescents and parents sometimes jointly | BSI and more | HIV | Terminal, incurable | Mean 14,7 | 4 years | New York City | USA | Improved. | Improved. | Improved. | High |
| Li, L. L., L., J.; Ji, G.; Wu, J.; Xiao, Y. | 2014 | Effect of a family intervention on psychological outcomes of children affected by parental HIV | AIDS & Behavior | Parents and children | Together for Empowerment Activities (TEA) family sessions and activities as well as and community events. | Self-esteem, parental distress and problem behavior | HIV | Chronic | 6 to 18 | 6 months | Village | China, Anhui province | Improved. | Improved. | Improved. | Moderate |
| Bröning, S., Sack, P-M.; Haevelmann, A.; et al | 2018 | A new preventive intervention for children of substance-abusing parents: Results of a randomized controlled trial | Child & Family Social Work | Children with info to parents | Psychoeducational preventive intervention "TRAMPOLINE" vs a non-educational "fun and play" group. | Knowledge, Self-efficacy and Self-concept, peer, family parent school relationship, quality of life, mental distress, social isolation, coping strategies. | Substance abuse or dependency | Current or within the last year. | 8 to 12 | 6 months | Outpatient counselling centres | Germany | Improved | Improved | Improved | High |
| Breslend, Nl.; Parent, J.; Forehand, R.; Peisch, V.; Compas, BE. | 2019 | Children of parents with a history of depression: The impact of a preventive intervention on youth social problems through reductions in internalizing problems | Development & Psychopathology | Family | Family group cognitive–behavioral (FGCB) preventive intervention vs a written information comparison condition. | Youth internalizing problems, youth social problems | Major Depressive Disorder (MDD | Chronic | 9 to 15 | 24 months | Psychiatry | USA | Improved | Improved | Improved | High |

(Continued)

**Table 1.** (Continued)

| Reference | Year conducted | Title | Journal | Population | Intervention | Child out-come | Parental illness | Illness stage | Age of children (yrs) | Follow up (mths) | Setting | Location | Child out-come internalizing symptoms | Child out-come externalizing symptoms | Child out-come Psycho-social function | Quality assessment |
|---|---|---|---|---|---|---|---|---|---|---|---|---|---|---|---|---|
| Schoenfelder, EN., Chronis-Tuscano, A.; Strickland, I.; Almirall, D., Stein, MA. | 2019 | Piloting a sequential, multiple assignment, randomized trial for mothers with attention-deficit/hyperactivity disorder and their at-risk young children | Journal of Child and Adolescent Psychopharmacology | Family | Mothers were randomized to stimulant medication (MSM) or behavioral parent training (BPT) | K-SADS-PL | ADHD (among both mother and child) | Chronic | 5 to 8 | 6 months | Psychiatry | USA | Improved | | | Moderate |
| May Aasebl. Hauken, MA, Pereira, M, Senneseth, M | 2018 | The Effects on Children's Anxiety and Quality of Life of a Psychoeducational Program for Families Living with Parental Cancer and Their Network A Randomized Controlled Trial Study | Cancer Nursing | Family | Cancer PEPSONE Program (CPP). Psychoeducation focusing on knowledge and discussion the family's expressed needs for social. | Quality of life: KINDL and anxiety: Revised Child Manifest Anxiety Scale | Cancer in one parent | Cancer, no specific state mentioned | 8 to 18 years | 3 months and 6 months | Oncology | Norway | Anxiety were not improved. Quality of life was partly improved. | | | Moderate |

**Table 2. Included studies using qualitative design.**

| Reference | Year conducted | Title | Journal | Population | Intervention | Parental illness | Illness stage | Age of children (yrs) | Follow up (mths) | Setting | Location | Child outcomes | Parental outcomes | Quality assesment |
|---|---|---|---|---|---|---|---|---|---|---|---|---|---|---|
| Semple, C. J. M., E. | 2013 | Family life when a parent is diagnosed with cancer: impact of a psychosocial intervention for young children | European Journal of Cancer Care | Parents and children | CLIMB Group intervention to provide education, normalize emotions, support communication and improve coping. | Canser | Mixed canser stage | 7 to 11 | Post intervention | Onchology | UK | Children had fantasies and misconceptions surrounding cancer. This psychological intervention (peer support) normalized their experience of parental cancer. It also improved children's understanding of cancer and equipped them with coping strategies. | Parents are often the gatekeeper to how children learn about parental cancer. Parents expressed a lack of confidence and skills in communicating with their children about cancer and stated the need for professional input. | Moderate |
| Landry-Dattee, N. B., D.; Roig, G.; Bouregba, A.; Delaigue-Cosset, M. F.; Dauchy, S. | 2016 | Telling the Truth. . . With Kindness: Retrospective Evaluation of 12 Years of Activity of a Support Group for Children and Their Parents with Cancer | Cancer Nursing | Parents and children. | Clinician lead meetings for parents and children | Canser | Mixed canser stage | Mean 11,03 | 1 to 12 years after intervention | Onchology | France | The children expressed more benefits: better understanding of the disease, reduction of symptoms, meeting similar others than they expected. | Parents expected to meet professionals who would help them speak about the disease to reduce children's symptoms and these expectations were largely satisfied. | Low |
| Holland, C. H., A.; Joubert, L.; McDermott, F.; Niski, M. D.; Thomson Salo, F.; Quinn, M. A. | 2017 | My Kite Will Fly: Improving Communication and Understanding in Young Children When a Mother Is Diagnosed with Life-Threatening Gynecological Cancer | Journal of Palliative Medicine | Parents and children | My Kite Will Fly | Canser | Acute, chronic and terminal | 3 to 12 | Before and after | Onchology | Australia | No "safe place" state of "guarded watchfulness," looking out for potential "threats." Worries about observing increasing physical and emotional distress in an ill parent, while also being concerned for the health of a surviving parent. In the worst-case scenario, children felt responsible for what they observed as happening to a dying parent, sometimes blam-ing themselves. Daughters sometimes worried about the possibility of developing cancer like their mother. Children benefit from being given definite simple family tasks and roles during parental treatments. | Parents were routinely concerned about the impact of a cancer diagnosis on their family roles. Tried to retain optimism and hope in the face of ongoing uncertainty and fear about the future. Felt the family offered a safe haven when other family members remained supportive and their roles and relationships endured. Agreed that when established family security is disrupted by cancer, the impact on a child must be addressed. Protected dependent children as a highest priority; particularly in a role as a single parent. | Low |

*(Continued)*

**Table 2.** (Continued)

| Reference | Year conducted | Title | Journal | Population | Intervention | Parental illness | Illness stage | Age of children (yrs) | Follow up (mths) | Setting | Location | Child outcomes | Parental outcomes | Quality assesment |
|---|---|---|---|---|---|---|---|---|---|---|---|---|---|---|
| Bugge, K. E. H., S.; Darbyshire, P. | 2008 | Children's experiences of participation in a family support program when their parent has incurable cancer | Cancer Nursing | Families | Family Support Program | Canser, terminal | Terminal, incurable | 6 to 16 | After, within 6 weeks | Onchology | Norway | The program helped the children to feel more secure; increased their knowledge and understanding; helped them become aware of their own role, their family's strengths, and whom they could approach for help; and helped them realize that it was good and helpful to talk about the illness situation. They needed to talk in private without having to think about other family members' reactions, but they also needed to be in dialogue with other family members. | No data | Moderate |
| Isobel, S. P., Danielle, Meehan, Felicity | 2017 | They are the children of our clients, they are our responsibility': A phenomenological evaluation of a school holiday program for children of adult clients of a mental health service | Advances in Mental Health | Parents and children | The school holiday program | Mental health | Chronic | 9 to 17 Mean 13 | Post intervention | Mental health | Australia | Escapism, the unexpected comfort of connection and fun in safe relationships. | Respite for their children, access to information about mental illness and their children's enjoyment. | Low |
| Pihkala, H. S., M.; Cederstrom, A. | 2012 | Children in Beardslee's family intervention: relieved by understanding of parental mental illness | International Journal of Social Psychiatry | Family | Beardslees family intervention | Mental health | Chronic | 6 to 17 | 3 to 11 months after intervention | Psychiatry | Sweden | Increased knowledge and more open communication about parental illness. Childrens' sense of relief. | Stronger as a parent and common with the children: Increased knowledge and more open communication about parental illness. Childrens' sense of relief. | High |
| Afzelius, M. P., L.; Ostman, M. | 2017 | Families living with parental mental illness and their experiences of family interventions | Journal of Psychiatric & Mental Health Nursing | Parents and children | Parent support groups, child support groups and family meetings. | Mental health | Chronic | 10 to 12 | Unclear | Psychiatry | Sweden | Using strategies to lead a normal life. Adjusting to the needs of the ill parents. Balancing one's own life and the demands of the parental mental illness. | Using strategies to lead a normal life. Adjusting to the needs of the ill parents. Concern for the child's needs and seeking support. | High |
| Wolpert, M. H., I.; Martin, A.; Fagin, L.; Cooklin, A. | 2015 | An exploration of the experience of attending the Kidstime programme for children with parents with enduring mental health issues | Clinical Child Psychology & Psychiatry | Parents and children | Kids Time: monthly psychosocial education for children and parents | Mental health | Chronic | 4 to 16 | Post intervention | Mental health | UK | Initial engagement, sharing with others, learning about mental health, opportunity for fun and impact on family relationships. | Initial engagement, sharing with others, learning about mental health, opportunity for fun and impact on family relationships. | Moderate |
| Trondsen, M. V. T., Aksel | 2014 | Communal Normalization in an Online Self-Help Group for Adolescents with a Mentally Ill Parent | Qualitative Health Research | Children of parents with mental illness (COPMI) | On-line chat room | Mental health | Chronic | 15 to 18 | Before and after | Internet | Norway | Recognizability (recognizing each other's similar experiences), openness (discussing issues that had been kept secret), and agency (retaining independent active steps toward plans and ambitions). | No data | Moderate |

(*Continued*)

**Table 2.** (Continued)

| Reference | Year conducted | Title | Journal | Population | Intervention | Parental illness | Illness stage | Age of children (yrs) | Follow up (mths) | Setting | Location | Child outcomes | Parental outcomes | Quality assesment |
|---|---|---|---|---|---|---|---|---|---|---|---|---|---|---|
| Pihkala, H. D.-B., N.; Sandlund, M. | 2017 | Talking about parental substance abuse with children: eight families' experiences of Beardslee's family intervention | Nordic Journal of Psychiatry | Parents and children | Beardsleed family intervention | Substance use disorder | Chronic | 4 to 15 | Six months after intervention | Psychiatry | Sweden | Good to have spoken out in the family, to be able to speak to their parents frankly, especially about feelings. Their worries about the parent had decreased. Information about the parent's treatment, diagnosis, about the heritability of alcohol abuse, but also details like what, where, with whom, and why the parent drank. How they could better stand up for their own desires, such as not wanting to be with the parent during a weekend if the parent drank alcohol. | Help to find words to explain their illness, a demanding task for the parents, often associated with shame and guilt. Shame decreased after they had broken the silence about the abuse and the parent felt relief. Many parents described how it was hard but necessary to hear what the children had said, and difficult to view themselves from the children's perspective. Parent's understanding of their children increased. | High |
| Templeton, L. | 2014 | Supporting families living with parental substance misuse: the M-PACT | Child & Family Social Work | Family groups | M-PACT, Moving Parents and Children together | Substance use disorder | Chronic | 8 to 18 | After | Community | UK | Engaging with M-PACT, shared experiences, understanding addiction, changes in communication, healthier and united families, and ending M-PACT. | Engaging with M-PACT, shared experiences, understanding addiction, changes in communication, healthier and united families, and ending M-PACT. | High |

*Child outcome*, *externalized symptoms*. Ten of 20 studies reported statistically significant improved levels of externalized symptoms for the children in the intervention groups. Three studies showed no effect on externalized symptoms.

*Child outcome*, *pro-social behavior*. Eight of the 20 studies reported pro-social outcomes for the child. These all reported statistically significant improved prosocial behavior in the child.

## Qualitative methodology studies

**Contextual factors in studies using qualitative methodology.**   Seven of the studies with qualitative design were conducted with families with parental mental health problems, including two in the context of parental drug abuse. Four qualitative studies concerned parental cancer. The children were in the age span of 3 to 17 years. All mental health studies concerned chronic parental disease. The cancer studies included mixed parental illness stages except one which specifically addressed terminal parental cancer. The studies were conducted in Europe and Australia.

**Intervention programme effects on child and parental outcomes.**   Content analysis resulted in four concepts important to children and parents and three additional concepts important to parents (Table 3).

## Intervention programme effects on child outcomes

A content analysis of the qualitative intervention outcomes for the children resulted in the following four clusters of outcomes:

**Increased knowledge and understanding.**   By getting information about the parent's illness, the child's misconceptions were dispelled, and a more accurate interpretation of observations was possible for the child. The child understood the impact of the illness on the family relations and became aware of their own role in the family.

**More open communication.**   Changes in communication made it possible to speak out. This was a gradual and interactive process. There were needs to be able to speak in different settings about what had previously been secret: in private with a professional, with peers, with the parents and the rest of the family, as well as one's extended network. To be able to speak within the family was essential, and this was made easier through communication in the other forums.

**Children's sense of relief.**   The children had experienced that there was no safe place to speak out before the intervention. They had been guarded and watchful and had worried

**Table 3.  Result of content analysis of outcomes in qualitative studies in parents and children.**

| Shared concepts | Children out-come | Parental out-come |
|---|---|---|
| Knowledge | Increased knowledge and understanding | Increased knowledge about their illness |
| Communication | More open communication | Changes in communication |
| Coping strategies | Better access to healthy coping strategies | Stronger as parent |
| Feelings | Children's sense of relief | Reduced feelings of shame and guilt |
| **Parent specific concepts** | | |
| | | Changes in their children's behavior |
| | | Parent's understanding of their own children increased |
| | | Respite |

about parental symptoms they observed. They even felt responsible for the situation and blamed themselves. They feared developing the same illness as their parents. After interventions they felt more secure and felt their experience had been normalized by the unexpected comfort of connection when meeting peers in similar circumstances.

**Better access to heathy coping strategies.** The interventions promoted improved agency (retaining independent active steps towards plans and goals), a better ability to stand up for their own desires and using strategies to lead a normal life. The children appreciated being allowed to shoulder defined, small, practical tasks for the family. They needed escapism with opportunities for fun, balancing the needs of their own lives and the needs of the ill parent. They learnt that they could ask for and receive help and whom they could approach.

In addition, there were descriptions on how to start and end an intervention and statements that there was a reduction in the symptoms of the child.

## Intervention programme effects on parental outcomes

A Content Analysis of the Qualitative Intervention Outcomes for the Parents Resulted in the Following Seven Clusters of Outcomes:

**Increased knowledge about their illnes.** Interventions conveyed knowledge about the parental illness and increased the parents' understanding of the illness.

*Changes in communication.* The parents found that the interventions let them meet professionals who helped them find words and phrases which were helpful when talking about the illness with their children. They also practiced the skill of communication during the intervention.

**Feelings of shame and guilt.** Parents expressed feelings of shame and guilt as well as an ongoing uncertainty and fear about the future development of the illness. They also expressed a lack of confidence in how to talk to their children about their illness. Sharing their situation, feelings and experiences with others reduced feelings of shame and guilt. When the parents had talked about their illness with their children, they felt relief. Participating in interventions provided opportunities for parents to observe their children's sense of relief.

**Stronger as a parent.** When the illness disrupted family security, parents felt that the illness' impact needed to be addressed. They were concerned about the impact of the illness on family roles and wished relationships to endure. They wished to remain in the role of a parent providing a safe, healthy, supportive, united and hopeful family. Both the ill parent and the other family members needed to adjust to the needs of the ill parent. Strategies were used to lead as normal a life as possible. The parent was a gatekeeper to how, when and in which context the child was to learn about the parental illness.

**Changes in children's behavior.** The parents observed changes in their children's behavior, and concern about these changes made the parents seek assistance with the highest urgency. They felt the need for professional input. They wished their children to understand the illness better in order to reduce the children's symptoms.

**Parents' understanding of their children.** Parents described how it was hard but necessary to hear what the children had to say and difficult to view themselves from the children's perspective, yet this communication increased their understanding of their children. The interventions opened the parents' eyes as to how the illness had affected the children, especially in the case of substance abuse.

**Respite.** The parents expressed relief in seeing their children have fun and enjoy activities and relationships during the intervention.

Furthermore, there were descriptions on how to start and end an intervention.

The comparative analysis of qualitative results in cancer and mental health care settings was limited by the amount of data available, but a basic difference of emphasis was discernable. Parents with cancer were seeking help to find words to talk about cancer with their children when they realized the illness disrupted family life. Parent with mental health problems described the same, but in addition, they struggled with feelings of shame and guilt and found it difficult but necessary to view themselves from the children's perspective and hear what the children had to say. Children of parents with cancer appreciated gaining knowledge about cancer and the treatment, and their worries decreased through open communication. The same was found among the children with parents with mental health and substance abuse, with the addition of an expressed relief in the truth finally being told in the family, in getting help to balance their own lives with the needs of the parent with mental illness, and in receiving help to find a way to better stand up for their own desires.

## Discussion

This systematic review was aimed at identifying evidence based/ evidence informed concepts found in effective interventions which inform children of their parent's illness in all health care settings globally. Included published articles report that interventions have been shown to improve the child's levels of internalized and externalized symptoms as well as increase the child's capacity for growth and the parents' capacity for positive parenting. This has been consistently shown in studies of children from families where the parents suffer from cancer, HIV, mental illness and substance abuse. This finding corresponds to earlier systematic reviews [6, 11, 17, 19, 20]. The interventions included in this review all relate to both the parent and the child, with differing emphases and in different settings: either family interventions with opportunities for individuals to be heard or peer-groups which include communication with the family. No studies reporting on interventions for parents with multimorbidity or efforts in the field of Primary Health Care were found.

Keeping in mind the Programme Theory model, which describes mechanisms by which interventions facilitate child health (Fig 2), it becomes apparent that outcomes in the quantitative studies mainly quantify intervention effects at the level of child internalized and externalized symptoms before and after the intervention. The qualitative studies describe the same but in shared concepts such as knowledge, communication, coping strategies and feelings. The qualitative studies also add specific concepts for the parents (changes in the children's behavior, increased parental understanding of their own children, and joy over their children's' respite) (Table 3).

### Helpful concepts found in effective interventions

In the process of this systematic review it became apparent that this field spans from universal human needs to specific individual and contextual considerations. On one hand, human beings have the same needs regardless of whatever illness they have contracted and the exact intervention setting. On the other hand, consideration of the specific, individual situation is crucial. Human beings have human needs in illness and in health. Parents remain parents also in illness. Transgenerational transmission of illness remains a central consideration for parents in all life situations. Illness and thereby threat to survival are an existential threat and increases, not decreases, the centrality of parenting. Parenting is a process that happens interactively between the parent and the child. Successful interventions therefore satisfy both the needs of the parent and those of the child. A vital need for parents is to have the opportunity to remain in a positive parental role even during times of illness. The corresponding need for the child is to be allowed to remain in the role of child, helped to stay safe and to grow. In previous

research [60] it has been shown that illness can reverse the roles and make the child take over parental responsibilities. The type of illness is of importance: All illnesses cause parental incapacity and shame but some more than others. The studies describing interventions in families with parental cancer describe higher parental awareness of their children's situation. Mental health problems, however, impair mental functions in a direct way and can be more detrimental to parental awareness and capacity. Substance abuse distorts the mental faculties and awareness, and in this setting directly points to increased parental awareness of the situation of the child as a difficult but important outcome. The threat of parental death is apparent in the thinking of all families with illness, and there is a need for health care professionals to address the risk of parental death. There are different settings which can be helpful, such as weekly meetings with a family or a peer group, camps for families or individuals, and chatting with peers on the Internet. When the providers of the intervention keep in mind the full Programme Theory model: mechanism by which interventions facilitate child's health (Fig 2), all settings can potentially have a positive effect.

## Strengths and limitations

Strengths of this review include the broad search in all settings globally from inception to November 2019 and the methodologically stringent work by all five authors. This results in a novel and innovative systematic overview of the research field. There is accumulating data showing effects on child symptom reduction, and via this review's content analysis, qualitative findings were synthesized into a coherent description of child and parent experiences of interventions. The trustworthiness [61] of this analysis is sufficient, as the aim of the study is broad, the sample specific, and we use an established method for doing a systematic review [24], the text analyzed are previous published results and the analysis strategy was stringent [57].

Limitations of this review reflect the lack of large effectiveness studies with outcome data on the children. Because of the diversity of outcomes published in the field so far, we were in this review unable to conduct a meta-analysis and specify the effect size of the symptom reduction in the children. Another serious limitation is that methodologically sound studies were only found in the fields of mental health, cancer and HIV care. It is therefore unknown what the exact situation is for parents and children in other health care fields.

## Implications and recommendations for future research

Implications of these results are mainly the imperative need in all settings to remember that many patients also are parents and that their children need information, advice and support. This information, advice and support must be provided in positive cooperation with the parent, and if professionals use knowledge presented in current research, parents in different settings have been shown to welcome it. For this to be implemented in more than just a few clinician-patient encounters, institutional implementation is needed [62].

This overview could be helpful background material in reflections, encouraged by law and policy, by professional health care staff in all settings globally. During such reflection one might find perspectives and possibilities for small, simple interventions in the clinical routines of one's own practice.

There is, furthermore, an urgent need for larger, well-conducted effectiveness studies in settings other than mental health, cancer and HIV, such as emergency care, neurology and primary health care.

## Conclusions

Thirty-two articles were included in this review, 21 quantitative and 11 qualitative and majority of them were rated as having high methodological quality. Most of the research was conducted in mental health, including substance abuse (n = 22), but there were also studies done in cancer care (n = 6) and HIV care (n = 4). The children were in the age span of 4 to 17 years. Most studies concerned chronic parental disease. The studies were conducted predominantly in the USA and Europe. All interventions relate to both the parent and the child, with differing emphases and in different settings.

Eight-teen of 20 studies using quantitative method showed a small to moderately positive statistically significant intervention effect on the child's level of internalized symptoms. Ten of these 20 studies reported statistically significant improved levels of externalized symptoms.

Content analysis of the results of studies employing qualitative methodology resulted in four concepts important to both children and parents in interventions (increased knowledge, more open communication, new coping strategies and changed feelings) and three additional concepts important to parents (observed changes in their children's behavior, the parent's increased understanding of their own child and the relief of respite).

In the reviewed literature there is evidence of mild to moderate effects on child symptoms as well as concepts worth noting when trying to bridge the still existing knowledge gaps. In further efforts the challenges of implementation as well as adaptation to differing clinical and personal situations appear key to address.

## Supporting information

**S1 Checklist. PRISMA 2009 checklist.**
(DOC)

**S1 Appendix. Search strategy.**
(DOCX)

**S1 Table.**
(XLSX)

**S2 Table.**
(XLSX)

## Acknowledgments

The searches for this review were conducted with the help of librarians at Karolinska Institutet, Sweden. Capio Sweden funded the first author's contribution to this study. Anna Dahland Kim proofread and improved the language of the manuscript.

## Author Contributions

**Conceptualization:** Charlotte Oja, Tobias Edbom, Anna Nager, Jörgen Månsson.

**Data curation:** Charlotte Oja.

**Formal analysis:** Charlotte Oja.

**Funding acquisition:** Solvig Ekblad.

**Investigation:** Charlotte Oja.

**Methodology:** Charlotte Oja, Tobias Edbom, Anna Nager, Jörgen Månsson, Solvig Ekblad.

**Project administration:** Solvig Ekblad.

**Supervision:** Tobias Edbom, Anna Nager, Jörgen Månsson, Solvig Ekblad.

**Validation:** Tobias Edbom, Anna Nager, Jörgen Månsson, Solvig Ekblad.

**Writing – original draft:** Charlotte Oja.

**Writing – review & editing:** Tobias Edbom, Anna Nager, Jörgen Månsson, Solvig Ekblad.

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
