## [Decision Letter · Decision Letter 0]

12 Mar 2020

PONE-D-19-28601

Informing Children of Their Parent's Illness: A systematic review of Intervention Programs with Child Outcomes in All Health Care Settings Globally from Inception to 2019

PLOS ONE

Dear Ms Ekblad,

Thank you for submitting your manuscript to PLOS ONE. After careful consideration, we feel that it has merit but does not fully meet PLOS ONE’s publication criteria as it currently stands. Therefore, we invite you to submit a revised version of the manuscript that addresses the points raised during the review process.

Academic Editor: The article was reviewed by two independent reviewers and this academic editor. It was an interesting review, not commonly seen and unaddressed topic. This manuscript is a comprehensive review of intervention programs aimed at informing children of their parent's illnesses. While all steps in the review were well described, some limitations should be acknowledged, such as the exclusion of one of the largest dabases usually included (PUBMED). The authors need to explain how the other databases compensate for PUBMED. Also, using the same wording in the objectives as in the conclusion would make it a clearer connection that both objectives and conclusions are aligned. 

We would appreciate receiving your revised manuscript by Apr 26 2020 11:59PM. To enhance the reproducibility of your results, we recommend that if applicable you deposit your laboratory protocols in protocols.io, where a protocol can be assigned its own identifier (DOI) such that it can be cited independently in the future. For instructions see: http://journals.plos.org/plosone/s/submission-guidelines#loc-laboratory-protocols

We look forward to receiving your revised manuscript.

Kind regards,

Abraham Salinas-Miranda

Academic Editor

PLOS ONE

Journal Requirements:

http://bora.uib.no/handle/1956/15934

https://www.tandfonline.com/doi/full/10.1080/02673843.2018.1548360

https://www.ncbi.nlm.nih.gov/pubmed?Cmd=ShowDetailView&Db=pubmed&TermToSearch=28505032

In your revision ensure you cite all your sources (including your own works), and quote or rephrase any duplicated text outside the methods section. Further consideration is dependent on these concerns being addressed."

3. Please ensure that your search is up-to-date, i.e. carried out in the last 12 months.

4. Please ensure that you include a title page within your main document. You should list all authors and all affiliations as per our author instructions and clearly indicate the corresponding author.

5.  Thank you for stating the following in the Financial Disclosure section 

"The authors received no specific funding for this work. Capio Sweden funded the first author's contribution to this study".

 We note that one or more of the authors have an affiliation to the commercial funders of this research study : 'Capio Sweden'.

Additional Editor Comments (if provided):

This manuscript is a comprehensive review of intervention programs aimed at informing children of their parent's illnesses. While all steps in the review were well described, some limitations should be acknowledged, such as the exclusion of one of the largest dabases usually included (PUBMED). The authors just need to explain how the other databases compensate for PUBMED. Also, using the same wording in the objectives as in the conclusion would make it a clearer connection that both objectives and conclusions are aligned.

Reviewers' comments:

Reviewer's Responses to Questions

**Comments to the Author**

1. Is the manuscript technically sound, and do the data support the conclusions?

Reviewer #1: Yes

Reviewer #2: Yes

Reviewer #3: Yes

2. Has the statistical analysis been performed appropriately and rigorously? 

Reviewer #1: Yes

Reviewer #2: I Don't Know

Reviewer #3: Yes

3. Have the authors made all data underlying the findings in their manuscript fully available?

Reviewer #1: Yes

Reviewer #2: Yes

Reviewer #3: Yes

4. Is the manuscript presented in an intelligible fashion and written in standard English?

Reviewer #1: Yes

Reviewer #2: Yes

Reviewer #3: Yes

5. Review Comments to the Author

Reviewer #1: The manuscript is well written. It is an important piece in its field and relevant in other fields too. The authors have fulfilled all research and publication ethics as demanded by the journal. The authors have also provided relevant data underlying the findings presented in the manuscript.

Reviewer #2: 1. The identified concepts found in the systematic should be reflected in the abstract's result section. This will be helpful to readers.

2. The conclusion does not reflect the goals of the study, this should be corrected.

Reviewer #3: 1. The attempt to do this kind of review was wonderful

2. The all steps are well defined

3. Page no. 5 the law and policy protecting and supporting child development is varying from country to country so the review should be restricted to Sweden not globally.

4. Page no 11 PROSPERO registering no should be given along with the website address (already mentioned)

5. Why PUBMED database was not included in the searches? It may give you more relevant studies.

6. Inclusion and exclusion Criteria very well defined.

7. Did the author try to get relevant data by contacting the authors of non English article?

6. PLOS authors have the option to publish the peer review history of their article (what does this mean?). If published, this will include your full peer review and any attached files.

Reviewer #1: None

Reviewer #2: No

Reviewer #3: No

---

## [Author Response · Author response to Decision Letter 0]

15 Apr 2020

Thank you for constructive comments which we have answered in the letter to the Academic Editor

---

## [Editor Report · Decision Letter 1]

12 May 2020

Informing Children of Their Parent's Illness: A systematic review of Intervention Programs with Child Outcomes in All Health Care Settings Globally from Inception to 2019

PONE-D-19-28601R1

Dear Dr. Ekblad,

We are pleased to inform you that your manuscript has been judged scientifically suitable for publication and will be formally accepted for publication once it complies with all outstanding technical requirements.

With kind regards,

Abraham Salinas-Miranda

Academic Editor

PLOS ONE

Additional Editor Comments (optional):

The authors have addressed the minor revisions requested by the reviewer and provided a revision with tracked changes. I find the revisions satisfactory and I see not reason to request additional revision. The authors conducted a systematic review of intervention programs aimed at informing children of their parent's illness across global settings. The review addresses multiple databases and noted the limitations in the literature. After examining the changes requested, this Academic Editor considers that the work can be published without additional delays.
---

## [Editor Report · Acceptance letter]

15 May 2020

PONE-D-19-28601R1 

Informing Children of Their Parent's Illness: A systematic review of Intervention Programs with Child Outcomes in All Health Care Settings Globally from Inception to 2019 

Dear Dr. Ekblad:

I am pleased to inform you that your manuscript has been deemed suitable for publication in PLOS ONE. Congratulations! Your manuscript is now with our production department. 

With kind regards,

on behalf of

Dr. Abraham Salinas-Miranda 

Academic Editor

PLOS ONE